# Circulating Endocannabinoids in Canine Cutaneous Mast Cell Tumor

**DOI:** 10.3390/ani14202986

**Published:** 2024-10-16

**Authors:** Valentina Rinaldi, Fabiana Piscitelli, Andrea Boari, Roberta Verde, Paolo Emidio Crisi, Tiziana Bisogno

**Affiliations:** 1Department of Veterinary Medicine, Veterinary Teaching Hospital, University of Teramo, 64100 Teramo, Italy; aboari@unite.it (A.B.); pecrisi@unite.it (P.E.C.); 2Institute of Biomolecular Chemistry (ICB)-CNR, 34, 80078 Pozzuoli, Italy; fpiscitelli@icb.cnr.it (F.P.); roberta.verde@icb.cnr.it (R.V.); 3Institute of Translational Pharmacology (IFT)-CNR, Via Fosso del Cavaliere 100, 00133 Rome, Italy

**Keywords:** MCT, endocannabinoid, dog, tumor

## Abstract

Cutaneous mast cell tumors (cMCTs) are a common type of tumor in dogs. This study measured blood concentrations of endocannabinoids, namely 2-arachidonoylglycerol (2-AG) and *N*-arachidonoylethanolamine (AEA), and AEA congeners, namely *N*-palmitoylethanolamine (PEA) and *N*-oleoylethanolamine (OEA), in 17 dogs with cMCTs and 11 healthy dogs. Dogs with cutaneous mast cell tumors had significantly higher levels of 2-AG and lower levels of AEA and PEA compared to healthy dogs, while OEA levels showed no difference. The ability of these endocannabinoids to serve as differentiators between healthy dogs and those with tumors was evaluated, and 2-AG was found to be a highly accurate (98%) indicator for identifying dogs with cMCTs. Levels of 2-AG above 52.75 pmol/mL had 94% sensitivity and 90% specificity in detecting the tumor. This is the first study to show that endocannabinoid levels are altered in dogs with mast cell tumors, indicating their potential use as biomarkers for diagnosing or monitoring this type of cancer in the future.

## 1. Introduction

A cutaneous mast cell tumor (cMCT) is the most common cutaneous neoplasia in dogs, accounting for approximately 7–21% of all skin tumors [1]. This tumor exhibits highly variable biological behavior, characterized by widespread metastases [1,2]. The clinical staging, according to the World Health Organization (0-IV WHO, [3]), and histological grading (Patnaik I-III and Kiupel Low-High) [4,5] of cMCTs have been established as the most important prognostic factors [1,4,5,6]. To predict the prognosis of canine mast cell tumors (MCTs), it is recommended to use a combined grading system, which classifies MCTs histologically into Grade I/Low, II/Low, II/High, or III/High categories [7]. Approximately 95% of Grade I/Low MCTs can be effectively treated with surgery alone. Despite advancements in the diagnosis and treatment of mast cell tumors in dogs, there remains a subset of patients for whom this condition proves fatal, as is often the case in dogs affected by Grade III/High MCTs, which have a median survival time of 108 days and a 1-year survival rate of 16% [8]. Although the influence of chronic skin inflammation and exposure to irritating compounds have been suggested [9], the underlying etiology of canine cMCTs remains incompletely understood, and deciphering the molecular mechanisms driving tumor development and progression is essential for advancing effective therapeutic strategies. Recent studies have shown the altered lipid metabolism in tumors, which supports their rapid growth, evasion of cell death, and metastasis [10]. *N*-arachidonoylethanolamine (anandamide, AEA) and 2-arachidonoylglycerol (2-AG) are endogenous lipids known as endocannabinoids (eCBs) [11]. AEA and 2-AG together with cannabinoid receptors (CB1 and CB2) and enzymes involved in eCBs’ biosynthesis and degradation are part of a complex lipid network known as the endocannabinoid system (ECS). Besides AEA, other *N*-acylethanolamines, such as *N*-palmitoylethanolamine (PEA) and *N*-oleoylethanolamine (OEA), are involved in the regulation of a wide range of biological functions [12,13]. The ECS is deeply involved in the maintenance of skin physiological conditions, and its dysregulation might contribute to several skin diseases, cancer included [14,15,16]. Moreover, dysregulation of the ECS has been associated with inflammatory conditions and cancers in both humans and dogs [17,18,19,20]. In veterinary small animal oncology, only one study showed that plasma levels of AEA, PEA, and OEA were increased and 2-AG levels were decreased in both B-cell and T-cell immunophenotype canine lymphomas, compared to in healthy control dogs [19]. Additionally, a recent study [21] showed that CB2 expression is conserved in canine diffuse B-cell lymphoma (DLBCL) but does not correlate with clinical outcome. Recently, the expression of CB1 and CB2 in cMCT was evaluated [22]. Notably, CB1 and CB2 are highly expressed only in low-grade Kiupel cMCT, and their immunoreactivity was inversely correlated with Kiupel grading [4]. To date, a potential dysregulation of eCB levels in canine cMCT has never been investigated. In the present study, the authors aimed to fill this gap by measuring plasma levels of AEA, 2-AG, OEA, and PEA in dogs affected by cMCT and in healthy control dogs. Additionally, the authors sought to explore whether these levels could be correlated with the clinical stage and histological grading of cMCT, thereby shedding light on their potential implications in tumor progression and prognosis.

## 2. Materials and Methods

### 2.1. Study Population

Dogs with a diagnosis of cMCT presented to the Veterinary Teaching Hospital (VTH) of the Department of Veterinary Medicine of University of Teramo, Italy, between February 2023 and June 2023, and were enrolled in this prospective observational cross-sectional study. For inclusion, all dogs without evidence of concurrent disease had to undergo the following: (1) physical examination, complete blood count (CBC), and serum biochemistry; (2) complete staging with cytological or histological evaluation of regional (RLN) or sentinel lymph nodes (SLN), thoracic radiographs, abdominal ultrasound, and fine needle aspiration (FNA) of the liver and spleen; and (3) histological diagnosis of cMCT according to the Patnaik and Kiupel system [4,5]. The clinical stage was classified according to WHO stages [3]. Dogs were excluded from the study if they were known to have received steroids, nonsteroidal anti-inflammatory drugs, and/or chemotherapy prior to blood collection in the six months preceding the study. Only dogs that had received treatment with antihistamines (i.e., cetirizine 5 mg/dog, oral administration, once a day; famotidine 2 mg/Kg SID oral administration, once a day) were allowed to be enrolled in this study. Eleven healthy dogs with a normal body condition score (BCS) of 5/9 who presented at Veterinary Teaching Hospitals for a routine check-up, for pre-anesthetic evaluation before neutering, or for blood donors’ selection were included in the control group. These patients had no clinical or pathological evidence of disease according to an unremarkable history, physical examination, and the results of CBC, serum biochemistry, and voided urine analysis. No medical treatment had been administered to this group of dogs in the preceding 6 months, except for regular treatments for the control of ectoparasites and/or vaccination. This study was performed with full informed consent provided by the owners and has been approved by the Animal Research and Ethics of the Universities in Chieti-Pescara, Teramo, and Experimental Zooprophylactic Institute of AeM (CEISA; UNICHD12 N. 1168).

### 2.2. Endocannabinoids Analysis

At the time of the visit at the VTH of Teramo, each dog included in the study underwent blood sampling after a 12 h overnight fasting period. To overcome the possible influence of the circadian rhythm in eCBs secretion, all samples were collected between 8:00 a.m. and 12:00 p.m. Blood was collected in Vacuette K3-EDTA tubes, immediately centrifugated for 10 min, 2000 rcf, at +4 °C, and the obtained plasma was immediately stored in 2 mL polypropylene tubes at −80 °C until analysis. Plasma samples were extracted in 5 volumes of chloroform/methanol (2:1), containing 5 pmol of d8-AEA, 50 pmol of d4-PEA, d2-OEA, and d5-2-AG (Cayman Chemicals, Ann Arbor, MI, USA). The lipid-containing organic phase was dried down in a rotating evaporator and pre-purified by open-bed chromatography on silica gel columns eluted with increasing concentrations of methanol in chloroform. Fractions eluted with chloroform/methanol 9:1 by vol. (containing AEA, 2-AG, OEA, and PEA) were collected, and aliquots were analyzed by isotope dilution–liquid chromatography/atmospheric pressure chemical ionization/mass spectrometry (LC-APCI–MS) using a Shimadzu high-performance liquid chromatography (HPLC) apparatus (LC-10ADVP, Shimadzu, Milan, Italy) coupled to a Shimadzu quadrupole mass spectrometer (LCMS-2020, Shimadzu) via a Shimadzu Atmospheric Pressure Chemical Ionization (APCI) interface (Shimadzu). LC analysis was performed in the isocratic mode using a Kinetex C18 column (Phenomenex, Torrance, CA, USA, 15 cm × 4.6 mm, 5 μm) and methanol/water/acetic acid (85:15:1 by vol.) as a mobile phase with a flow rate of 1 mL/min. MS detection was carried out in the selected ion monitoring mode using *m*/*z* values of 356 and 348 (molecular ion +1 for deuterated and undeuterated AEA), 384.35 and 379.35 (molecular ion +1 for deuterated and undeuterated 2-AG), 304 and 300 (molecular ion +1 for deuterated and undeuterated PEA), and 328 and 326 (molecular ion +1 for deuterated and undeuterated OEA). In the case of 2-AG, the data represent the combined signals from the 2- and 1(3)-isomers since the latter are most likely generated from the former via acyl migration from the sn-2 to the sn-1 or sn-3 position. The levels of eCBs were then calculated based on their area ratios with the internal deuterated standard signal areas, and their amounts were expressed as pmol/mL of plasma.

### 2.3. Statistical Analysis

Data analysis was performed using statistical software (GraphPad Prism version 6.01, GraphPad Software, La Jolla, CA, USA, www.graphpad.com); for multiple linear regression analysis, data were processed and analyzed using R version 4.4.1. (R Core Team, Vienna, Austria). All data were evaluated using standard descriptive statistics and reported as mean ± standard error of mean (SEM) or as median and range (minimum–maximum) depending on the data’s distribution. Normality was checked using the D’Agostino Pearson test. Comparisons between the two groups (i.e., control vs cMCT; clinical stage I and II vs. clinical stage III; received antihistaminic treatment vs. did not receive antihistaminic treatment; size of tumors > 3 cm vs. size of tumors < 3 cm; presence of ulceration vs. absence of ulceration) were performed using the unpaired *t*-test or the Mann–Whitney test.

To assess the potential influence of age and BCS on circulating levels of eCBs, a multiple linear regression analysis was performed. Age and BCS were included as independent variables, while levels of AEA, 2-AG, PEA, and OEA were the dependent variables. The models were used to adjust for potential confounding effects of these covariates on eCB levels.

For those eCBs that showed statistically significant differences in the comparison between the control and cMCT groups, the sensitivity (Se), specificity (Sp), and negative and positive likelihood ratios (−LR and +LR) at different cut-off points and receiver operating characteristic (ROC) curves were used to assess the accuracy of the eCBs in distinguishing dogs affected by cMCT from healthy dogs. A *p* value < 0.05 was considered significant; a 95% confidence interval (CI) was also reported.

## 3. Results

### 3.1. Patient Population

Seventeen dogs with cMCTs were included in this study, with the dogs having a mean age of 8.6 years (SEM ± 0.6), a mean body weight of 18.0 Kg (SEM ± 2.2), and a mean BCS of 5.9 (SEM ± 0.3). In three cases, cMCT was diagnosed as high-grade (III grade Patnaik, High Kiupel), while in the remaining fourteen cases, low-grade cMCT was diagnosed (II grade Patnaik, Low Kiupel). Seven dogs were in WHO stage I, four dogs were in WHO stage II, and six dogs were in WHO stage III, while no dogs were in WHO stage IV. Five dogs had a tumor size of >3 cm, and three of these were high-grade at the histological exam. Five dogs showed ulcers, and two of these were high-grade. Four dogs received antihistamine therapy before the visit. Population characteristics are summarized in Table 1. Healthy dogs (n = 11) enrolled in the control group (Appendix A) had a lower age (mean 5.5 years; SEM ± 0.7; *p* = 0.0048; 95% CI −5.070 to −1.016) and BCS (mean 5.0; SEM ± 0.0; *p* = 0.014; 95% CI −1.576 to −0.1889), while no differences were observed in body weight (mean 22.2 kg; SEM ± 3.4; *p* = 0.30; CI −3.899 to 12.14) compared to cMCT dogs. The most represented breed was cross breed.

### 3.2. Endocannabinoids and Related NAEs in Canine Plasma Samples

Multiple linear regression analysis revealed no statistically significant associations between age or BCS and circulating levels of AEA, 2-AG, PEA, or OEA. For AEA, neither age (*p* = 0.124) nor BCS (*p* = 0.817) significantly impacted its levels. Similarly, 2-AG levels were not significantly associated with age (*p* = 0.369) or BCS (*p* = 0.212). Analysis of PEA and OEA also showed no significant effects of age (*p* = 0.773 and *p* = 0.424, respectively) or BCS (*p* = 0.993 and *p* = 0.463, respectively). Thus, age and BCS were not major determinants of circulating eCB levels in the cohort studied.

No significant differences (*p* > 0.05) were found in the levels of AEA, 2-AG, PEA, and OEA between the early stages (I and II) and the more advanced stage (III) in dogs with cMCTs, between dogs treated with antihistamines versus those not treated, between dogs with larger (>3 cm) versus smaller (<3 cm) tumors, and based on the presence or absence of ulceration (Appendix A).

Dogs affected by cMCT (n = 17), regardless of the WHO stage and histologic grading, had higher plasma levels of 2-AG (mean 141.1; SEM ± 77.5; *p* = 0.0001; 95% CI 58.18 to 156.8) and lower levels of AEA (mean 5.7; SEM ± 3.5; *p* = 0.0012; 95% CI −6.835 to −1.909) and PEA (mean 30.1; SEM ± 5.3; *p* = 0.0075; 95% CI −8.756 to −1.492) compared to the control group. No differences were observed at the OEA level between healthy and cMCT dogs (mean 18.9; SEM ± 8.5; *p* = 0.9264; 95% CI −5.886 to 6.445) (Figure 1). The ability of eCBs to help discriminate between healthy and cMCT dogs was interrogated through the area under the ROC curve (AUC) (Figure 2). Accuracies of 0.98 (95% confidence interval [CI], 0.94–1.02) for 2-AG, of 0.85 (95% CI, 0.71–0.99) for AEA, and of 0.81% for PEA (95% CI, 0.64–0.69) were found. Values > 52.75 pmol/mL of 2-AG showed 94% sensitivity and 90% specificity in distinguishing cMCT (Table 2; Figure 2).

## 4. Discussion

In the recent scientific literature within the field of veterinary oncology, there has been an increasing emphasis on the research and utilization of biomarkers. Despite MCTs being among the two most common tumors in dogs, studies on biomarker identification have been limited, and to date, no biomarkers have been characterized that can be obtained from blood samples [23,24,25]. The only predictive marker for which research has been published to date in canine MCT is c-KIT mutational status [26]. c-KIT mutations are a tissue marker that can indicate more aggressive forms of MCT, can predict outcomes, including the likelihood of recurrence or metastasis, and can guide treatment choices [27].

Recently, Rinaldi and co-authors reported that CB1 and CB2 are expressed in cMCTs and are inversely correlated with tumor grade [22]. The current study expanded this research from cannabinoid receptors to eCBs, endogenous ligands able to bind to and functionally activate cannabinoid receptors. Although in recent years the circulating levels of eCBs have been determined in canine chronic enteropathy [18] and lymphoma [19], there are no data regarding the circulation of eCBs in the context of MCTs in dogs. In this study population, past authors have evaluated WHO clinical stage [3], histological grading [4,5], and clinical characteristics like ulcers and size [1]. In agreement with what has been reported previously in the literature [1,28], the size (i.e., >3 cm) and/or the presence of ulceration were characterized by more aggressive biological behavior except for in one case, where an ulcer was present in a low-grade cMCT in clinical stage I.

In this study, the authors observed a significant increase in 2-AG in dogs affected by cMCT compared to the control group. The increase in 2-AG levels observed in this study is in agreement with the alteration of eCB levels highlighted in other types of tumors. Indeed, 2-AG was increased in human tissues in endometrial carcinoma [29], colorectal cancer [30], pituitary adenomas [31], glioma [32], and hepatocellular carcinoma [33]. Additionally, the authors unveiled a reduction in AEA content in the plasma of MCT patients, clearly providing evidence in favor of the distinct involvement of the two main eCBs in the disease. Furthermore, it should also be mentioned that an inverse regulation, characterized by increased 2-AG and decreased AEA levels, was observed in the human plasma of 304 cancer patients [20], as well as in human tissue glioma [32].

In veterinary medicine, the only other study on eCBs in canine neoplasia showed a reduction in 2-AG and an increase in AEA serum levels in dogs affected by DLBCL [19], with these results being in opposition to the alteration of eCB levels highlighted in human patients suffering from the same tumor [34].

Although the molecular mechanism underlying the alteration of eCBs was not investigated in this study, it is possible to hypothesize that dysregulations of eCBs’ biosynthetic and/or degradative processes could account for the changes in eCB levels reported here. For instance, the upregulation of diacylglycerol lipase (DAGL) or the downregulation of monoacylglycerol lipase (MAGL) could be consistent with the increase in 2-AG reported in the tissue in hepatocellular carcinoma [33] and in endometrial carcinoma [29], respectively.

In order to critically evaluate their diagnostic potential in cMCT, the remarkable accuracy of eCBs in distinguishing between control and cMCT dogs was thoroughly investigated. Notably, 2-AG demonstrated an exceptional ability to help discriminate between healthy and cMCT dogs, as evidenced by an impressive area under the ROC curve (AUC) of 0.98, highlighting its strong potential as a biomarker for cMCT detection. Similarly, the significant decrease in AEA levels in cMCT dogs, with an AUC of 0.85, further supports its promise as a diagnostic marker for cMCT. Further investigations are warranted to determine whether similar accuracy can be observed between plasma eCB levels and different groups of other spontaneous tumors in dogs. Moreover, in the present study, we also observed that PEA levels in the plasma of dogs affected by MCT were lower compared to healthy dogs, with an impressive area under the ROC curve (AUC) of 0.81. The trend towards a reduction in AEA and PEA plasma levels in cMCT dogs appears to be in agreement with the well-documented observation that AEA shares with PEA a common biosynthetic pathway, and suggests that a comprehension of the alteration of specific NAEs-mediated signaling could contribute to the unveiling of pathophysiological mechanisms of cMCT. Despite its structural similarity to AEA, PEA does not bind to CBs, and its therapeutic effects are mainly based on the activation of other molecular targets [35]. Of note, we showed that none of the compounds quantified in this study, PEA included, seem to be affected by treatment with antihistamines. Indeed, four dogs had received antihistamine therapy before the visit, but neither cetirizine nor famotidine oral administration significantly impacted eCB levels. PEA is extensively studied for its anti-inflammatory effects, particularly for its role in attenuating the degranulation of mast cells (MCs) triggered by substance P [36]. Moreover, PEA has been shown to prevent IgE-induced degranulation in isolated canine skin MCs (histamine, PGD2, and TNF-α release). The decrease in PEA may suggest a dysfunction within the intrinsic response mechanism intended to moderate the inflammatory reactions of MCs [37]. The mechanisms underlying its anti-inflammatory effects are mainly due to its ability to bind the peroxisome proliferator-activated receptor α (PPAR-α) [38] that also mediates its effect in several carcinogenesis mechanisms [39,40]. Moreover, PEA modulates AEA effects by increasing AEA endogenous levels and potentiating its binding at CB1 and CB2 receptors or at transient receptor potential vanilloid type-1 (TRPV1) channels, by downregulation of fatty acid amide hydrolase (FAAH), the enzyme responsible of AEA inactivation, or by a positive allosteric modulation of TRPV1 channels [35].

Finally, even though it has been previously reported that low concentrations of OEA promote the migration of neoplastic cells through the activation of Erk signaling pathways [41] and that high concentrations inhibit tumor cell migration [42,43], in this study, no differences were reported in OEA levels in cMCT dogs compared to healthy dogs. This supports a different involvement of N-acylethanolamines and their receptor targets in cMCT, as OEA is not a ligand at classical CB receptors.

Despite the insights gained from this study, several limitations should be acknowledged. Firstly, the sample size was relatively small, patients in the IV clinical stage were not enrolled, and a limited number of dogs affected by high-grade cMCT was included. Moreover, the study was conducted using a cross-sectional design, and samples were not matched for age and BCS, although these factors do not appear to affect the results regarding circulating eCBs [18]. Based on the above-mentioned considerations, it seems evident that future studies are needed to understand the molecular mechanism underlying the alterations of eCBs observed in this study. In particular, it will be necessary to assess if the dysregulation of eCBs’ metabolism in terms of expression, and/or the activity of biosynthetic (DAGL and NAPE) and/or degradative (FAAH and MAGL) enzymes could account for the changes in eCB levels reported here. It will be important to evaluate whether systemic changes in the different elements of the ECS are the mirror image of those in bioptic tissues. Finally, longitudinal studies are also required to ascertain whether eCB tone and signaling undergo specific alterations during the different clinical stages of MCTs, and whether they are associated with disease progression or prognosis. On the other hand, prospective studies are needed to evaluate the modulation of eCBs’ plasmatic tone as a potential examination of the efficacy of pharmacological therapy. Therefore, it is possible to foresee that a complete knowledge of the action and function of the ECS in cutaneous neoplastic diseases might lead to the development of alternative therapeutic strategies based on the pharmacological modulation of the ECS. Indeed, synthetic pharmacological tolls such as receptor agonists and/or inhibitors of eCB hydrolytic enzymes, as well as antagonists and/or inhibitors of eCB biosynthetic enzymes, might enhance or reduce, respectively, the activity of the ECS. On this basis, it will be possible to hypothesize their use in the fine-tuning of the ECS’s therapeutic potential in MCTs. Lastly, data reported here, together with previous evidence that cannabinoid receptors are expressed in cMCT [22] and the above-suggested future studies, can provide the biochemical and molecular basis to justify the increasing interest and use of cannabinoids, particularly CBD products, in veterinary medicine [44,45].

## 5. Conclusions

In conclusion, to the best of authors’ knowledge, this study revealed for the first time alterations of circulating eCBs in the plasma profile in cMCTs. These findings, together with previous evidence that cannabinoid receptors are expressed in cMCT [22], support the role of the ECS in the pathogenesis of cMCT, paving the way for future cannabis-based therapeutic strategies in the veterinary oncology field.

## Figures and Tables

**Figure 1 animals-14-02986-f001:**
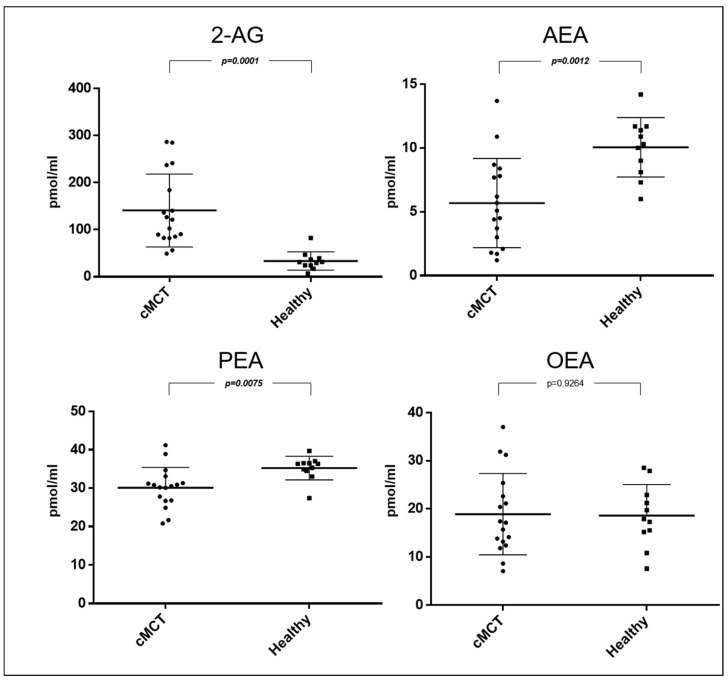
Scatter dot plot of plasmatic concentrations expressed in pmol/mL of 2-arachidonoylglycerol (2-AG), *N*-arachidonoylethanolamine (AEA), *N*-palmitoylethanolamine (PEA), and N-oleoylethanolamine (OEA) in dogs affected by cutaneous mast cell tumor (cMCT, n = 17) and in healthy dogs (n = 11). Bold font denotes statistical significance.

**Figure 2 animals-14-02986-f002:**
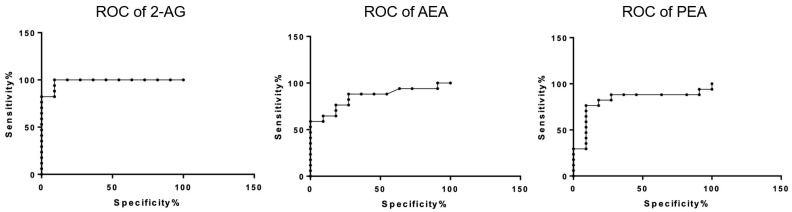
Receiver operating characteristic (ROC) curves of 2-arachidonoylglycerol (2-AG), *N*-arachidonoylethanolamine (AEA), and *N*-palmitoylethanolamine (PEA).

**Table 1 animals-14-02986-t001:** Study population.

#	Breed	Sex	Age (Years)	Body Weight (Kg)	BCS	Histology	Antihistaminic Treatment	WHO Stage	Ulcer	Size
1	Cross breed	M	13	25	6	MCT high-grade	No	II	Yes	>3 cm
2	Labrador Retriever	F	5	27	5	MCT low-grade	No	I	No	<3 cm
3	Cross breed	NM	10	25	7	MCT low-grade	No	III	No	<3 cm
4	Boxer	M	9	31	5	MCT low-grade	No	I	No	<3 cm
5	Cross breed	M	10	20	8	MCT low-grade	No	I	No	<3 cm
6	Cross breed	F	7	18	5	MCT low-grade	No	III	Yes	>3 cm
7	Cross breed	SF	10	23	5	MCT low-grade	No	I	No	<3 cm
8	Cross breed	SF	11	12	5	MCT high-grade	Yes	II	No	<3 cm
9	Cross breed	SF	7	5	6	MCT low-grade	No	I	No	<3 cm
10	Poodle	SF	6	3	5	MCT low-grade	Yes	III	No	<3 cm
11	Shar-pei	NM	5	30	6	MCT high-grade	No	II	Yes	>3 cm
12	Pug	SF	13	9	8	MCT low-grade	No	III	Yes	<3 cm
13	Galgo	SF	10	22	5	MCT low-grade	Yes	I	No	<3 cm
14	Cross breed	SF	6	7	5	MCT low-grade	No	II	No	<3 cm
15	Cross breed	NM	5	19	5	MCT low-grade	Yes	III	No	<3 cm
16	English Setter	NM	10	25	7	MCT low-grade	No	III	No	<3 cm
17	Maltese	M	9	6	7	MCT low-grade	No	I	Yes	<3 cm

Abbreviations: M: Male; NM: Neutered Male; F: Female; SF: Spayed Female.

**Table 2 animals-14-02986-t002:** Results of the receiver operating characteristic (ROC) curve analysis.

	2-AG	AEA	PEA
Area	0.9840	0.8529	0.8182
Std. Error	0.01921	0.07246	0.08843
95% Confidence Interval	0.9463 to 1.022	0.7109 to 0.9950	0.6448 to 0.9915
*p* Value	<0.0001	0.001916	0.005147
Value	>52.75	<6.750	<32.15
Sensitivity (%)	94.12	64.71	76.47
95% CI	71.31% to 99.85%	38.33% to 85.79%	50.10% to 93.19%
Specificity (%)	90.91	90.91	90.91
95% CI	58.72% to 99.77%	58.72% to 99.77%	58.72% to 99.77%
Likelihood Ratio	10.35	7.118	8.412

## Data Availability

The data that support the findings of this study are available on request.

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
