# Peer review of "Circulating Endocannabinoids in Canine Cutaneous Mast Cell Tumor"

_animals, 2024, doi:10.3390/ani14202986_

Round 1
Reviewer 1 Report
Comments and Suggestions for Authors
The assigned manuscript describes a study that is framed in the search for possible biomarkers for early detection of cancer, which is very important in human medicine and also in veterinary medicine, although in the latter it is still in its beginning stages.
Precisely for this reason, studies centered on noninvasive diagnostic methods such as detection by biomarkes are very interesting, especially those that outline for the first time the detection of a biomarker in a tumor and, all the more so, in a tumor so frequent in dogs-a truly daily problem for veterinary oncologists.
The present study, moves from a single veterinary scientific contribution on the detection of endocannabinoids in canine lymphomas and aims to evaluate the possible use of these biomarkers in the field of dog mast cell tumors.
The manuscript is well written, the experimental design clear and the results very interesting and commented extensively in the discussion
The work is therefore very good and my rating is therefore “minor revision.”
Among these revisions one needs special consideration by the authors: the fact that some subjects had been given antihistamines. This treatment does not seem to have affected the results. It is worth mentioning in the discussion and stressing that this does not seem to have affected the results obtained.
Another review, which is minimal, concerns lines 45-46, in the introduction: mast cell tumor is a tumor that in dogs has been the subject of a great number of studies that have led to very good survival results, so I would not be so drastic in those lines and cite more recent literature.
One could say that...although there have been numerous advances in the diagnosis and treatment of canine mastocytoma, this tumor is still vontinues to be fatal in a ..... percentage of cases.
Author Response
Dear Editor and dear Reviewers
Please find enclosed a fully revised version of our submitted article. We have done our best to comply with nearly all the several comments of the handling Reviewers, whom we thank for their constructive suggestions. The changes made to the manuscript are described point-to-point below. We believe that the article has been much improved by these changes and hope that it will be now deemed acceptable for publication.
Reviewer 1
The assigned manuscript describes a study that is framed in the search for possible biomarkers for early detection of cancer, which is very important in human medicine and also in veterinary medicine, although in the latter it is still in its beginning stages.
Precisely for this reason, studies centered on noninvasive diagnostic methods such as detection by biomarkes are very interesting, especially those that outline for the first time the detection of a biomarker in a tumor and, all the more so, in a tumor so frequent in dogs-a truly daily problem for veterinary oncologists.
The present study moves from a single veterinary scientific contribution on the detection of endocannabinoids in canine lymphomas and aims to evaluate the possible use of these biomarkers in the field of dog mast cell tumors.
The manuscript is well written, the experimental design clear and the results very interesting and commented extensively in the discussion
The work is therefore very good and my rating is therefore “minor revision.”
Among these revisions one needs special consideration by the authors: the fact that some subjects had been given antihistamines. This treatment does not seem to have affected the results. It is worth mentioning in the discussion and stressing that this does not seem to have affected the results obtained.
Reply: Dear Reviewer, thank you for your appreciation and for your valuable comments. We have added a sentence in the Discussion section regarding the results related to the use of antihistamines.
Another review, which is minimal, concerns lines 45-46, in the introduction: mast cell tumor is a tumor that in dogs has been the subject of a great number of studies that have led to very good survival results, so I would not be so drastic in those lines and cite more recent literature.
One could say that...although there have been numerous advances in the diagnosis and treatment of canine mastocytoma, this tumor is still vontinues to be fatal in a ..... percentage of cases.
Reply: Thank you for your comment, we have added a new part in the introduction.
Reviewer 2 Report
Comments and Suggestions for Authors
In this study, Rinaldi et al aimed to characterize the plasma levels of N-arachidonoylethanolamine (AEA), 2-arachidonoylglycerol (2-AG), N-palmitoylethanolamine (PEA) and N-oleylethanolamine (OEA) in 17 dogs with cutaneous mast cell tumors, as compared to 11 healthy dogs. The authors identified statistically significant higher levels of 2-AG and lower levels of AEA and PEA as compared to heathy dogs, that were able to discriminate between the two conditions with an AUC>0.8. I have the following suggestions for the authors:
Major comments
-Lines 74-77 and 137-140: This study does not show any of these correlations/comparisons. Unless the authors show correlations of these lipids with tumor stage, grade receipt of medication tumor size or presence of ulceration, these statements should be removed.
-Lines 158-159 indicate that healthy dogs had a statistically significant lower age and BCS, and in this case data should be adjusted for age and BCS before proceeding further. Was this analysis performed?
-There are some studies that show sex-related differences in endocannabinoid pharmacology and metabolism (as summarized here: Sex differences in cannabinoid pharmacology: A reflection of differences in the endocannabinoid system? - PMC (nih.gov)). I would encourage the authors to look into potential differences between the prognostic validity of these biomarkers in male and female dogs separately and discuss this issue in the Discussion section.
-The authors are encouraged to make a diagram showing a potential mechanism of eCB-related tumorigenicity, based on the findings of this study and their previous one assessing the levels of CB1 and CB2 in cMCT.
-Lines 275-277: Please explain in the Discussion section how these two studies pave the way for cannabis-based therapeutics. This could be included in the diagram above.
-Please provide specific suggestions for future studies.
Minor comments
-Table 1 and Suppl. Table 1: Please define NM and SF
-Lines 268-269: "Again...circulating eCBs". Do the authors have a reference for this statement? If not, please remove.
Comments on the Quality of English Language
-line 49: "and exposure to irritating compound have been evocated": Please rephrase this statement
-line 83: "must underwent": Please rephrase
-line 211: "it should also mentioned" to "it should also be mentioned"
-Line 270: "are needed to assess... occurs": Please rephrase
Author Response
Dear Editor and dear Reviewers
Please find enclosed a fully revised version of our submitted article. We have done our best to comply with nearly all the several comments of the handling Reviewers, whom we thank for their constructive suggestions. The changes made to the manuscript are described point-to-point below. We believe that the article has been much improved by these changes and hope that it will be now deemed acceptable for publication.
Reviewer 2
In this study, Rinaldi et al aimed to characterize the plasma levels of N-arachidonoylethanolamine (AEA), 2-arachidonoylglycerol (2-AG), N-palmitoylethanolamine (PEA) and N-oleylethanolamine (OEA) in 17 dogs with cutaneous mast cell tumors, as compared to 11 healthy dogs. The authors identified statistically significant higher levels of 2-AG and lower levels of AEA and PEA as compared to heathy dogs, that were able to discriminate between the two conditions with an AUC>0.8. I have the following suggestions for the authors:
Major comments
-Lines 74-77 and 137-140: This study does not show any of these correlations/comparisons. Unless the authors show correlations of these lipids with tumor stage, grade receipt of medication tumor size or presence of ulceration, these statements should be removed.
Reply: Dear Reviewer, thank you for your comments. We have incorporated the correlations made between the clinical aspects and the analysis results.
-Lines 158-159 indicate that healthy dogs had a statistically significant lower age and BCS, and in this case data should be adjusted for age and BCS before proceeding further. Was this analysis performed?
Reply: Thank you for your suggestion, we have added a section about these features.
-There are some studies that show sex-related differences in endocannabinoid pharmacology and metabolism (as summarized here: Sex differences in cannabinoid pharmacology: A reflection of differences in the endocannabinoid system? - PMC (nih.gov)). I would encourage the authors to look into potential differences between the prognostic validity of these biomarkers in male and female dogs separately and discuss this issue in the Discussion section.
Reply: Dear Reviewer, although the article you suggested presents data in both human and murine species, to date, no significant differences have been observed in dogs, as already published by this research group in 2021 (Febo et al.). This is likely due, in part, to the fact that dogs have a different hormonal profile compared to humans. Furthermore, many of the female and male subjects were neutered, which also impacts their hormonal balance.
-The authors are encouraged to make a diagram showing a potential mechanism of eCB-related tumorigenicity, based on the findings of this study and their previous one assessing the levels of CB1 and CB2 in cMCT.
Reply: Thank you for your comment. We agree with the reviewer that a graphical description of the molecular mechanisms underlying the variations observed in this study could be extremely appealing for the reader. On the other hand, the data from this study even when combined with those from the paper on CB1-CB2 receptors, are not sufficient to make a diagram showing the role of ECS in the tumorigenicity of mast cell tumors. Given the information we have, a graphical representation may be too speculative. For this reason, as you suggested, we described the future studies necessary to draw it.
-Lines 275-277: Please explain in the Discussion section how these two studies pave the way for cannabis-based therapeutics. This could be included in the diagram above.
-Please provide specific suggestions for future studies.
Reply: Thank you for your comment, we have added a specific suggestions for the future studies.
Minor comments
-Table 1 and Suppl. Table 1: Please define NM and SF
Reply:
Thank you for your comment, we have added the abbreviation
-Lines 268-269: "Again...circulating eCBs". Do the authors have a reference for this statement? If not, please remove.
Reply: Thank you for your comment. We have added the reference we were referring to ref 18 (Febo et al. 2021).
Comments on the Quality of English Language
-line 49: "and exposure to irritating compound have been evocated": Please rephrase this statement
-line 83: "must underwent": Please rephrase
-line 211: "it should also mentioned" to "it should also be mentioned"
-Line 270: "are needed to assess... occurs": Please rephrase
Reply:Thank you for your comment. We have modified the sentences.
Round 2
Reviewer 2 Report
Comments and Suggestions for Authors
I would like to thank the authors for addressing my comments and improving the manuscript, that is now suitable for publication.